# LLM aided semi-supervision for Extractive Dialog Summarization

**Nishant Mishra**[1,2*]    **Gaurav Sahu**[3,4]    **Iacer Calixto**[1,2]
**Ameen Abu-Hanna**[1,2]    **Issam H. Laradji**[4,5]

[1]Amsterdam UMC, Department of Medical Informatics, University of Amsterdam
[2]Amsterdam Public Health, Methodology, Amsterdam, The Netherlands
[3]University of Waterloo  [4]ServiceNow Research
[5]University of British Columbia

## Abstract

Generating high-quality summaries for chat dialogs often requires large labeled datasets. We propose a method to efficiently use unlabeled data for extractive summarization of customer-agent dialogs. In our method, we frame summarization as a question-answering problem and use state-of-the-art large language models (LLMs) to generate pseudo-labels for a dialog. We then use these pseudo-labels to fine-tune a chat summarization model, effectively transferring knowledge from the large LLM into a smaller specialized model. We demonstrate our method on the TWEETSUMM dataset, and show that using 10% of the original labelled data set we can achieve 65.9/57.0/61.0 ROUGE-1/-2/-L, whereas the current state-of-the-art trained on the entire training data set obtains 65.16/55.81/64.37 ROUGE-1/-2/-L. In other words, in the worst case (i.e., ROUGE-L) we still effectively retain 94.7% of the performance while using only 10% of the data.

## 1 Introduction

Customer support chats are gaining popularity as a primary medium for servicing clients in industry. Summarization models that robustly capture critical information are therefore extremely beneficial for companies to support various downstream activities such as record-keeping, training. Unfortunately, standard supervised training of these models rely on large labeled datasets, which can be prohibitively expensive to build. While unsupervised summarization techniques exist (Zou et al., 2021; Zhang et al., 2021; Shang et al., 2018), enforcing the summary quality and style is still an ongoing challenge.

In general terms, chat summarization can be *abstractive*, where the generated summary is unconstrained(Goo and Chen, 2018; Chen and Yang, 2021; Gupta and Gupta, 2019), or *extractive*, where

the summary consists of parts of the original input (Feigenblat et al., 2021; Liu, 2019). In this work, we propose a novel *extractive dialog summarization* method based on a semi-supervised learning paradigm, and demonstrate that we either improve or perform comparably to the current state-of-the-art on the TWEETSUMM dataset (Feigenblat et al., 2021), **while using only 10% of the training data**.

Pseudolabeling(Lee, 2013; Pham et al., 2021) is a popular and efficient method for semi-supervised learning. Pre-trained LLMs encode a vast breadth of knowledge in various tasks and can generate human like response. They are good annotators(Wang et al., 2021; Ding et al., 2023) and also ideal for transferring knowledge through distillation(Kim et al., 2022; Kim and Rush, 2016; Liu et al., 2021). Hence, we decided to use LLMs 1) as *weak labellers* to generate automatic summaries for a large set of unlabelled examples (i.e., pseudo-labels); and 2) as *evaluators* to choose only a small number of high-quality (pseudo-labelled) examples to include in the next training cycle. We report strong results on the TWEETSUMM dataset in zero- and few-shot semi-supervised settings. Our main contributions are as follows.

- We introduce a semi-supervised method for extractive summarization to distill knowledge from general-purpose LLMs into smaller specialized summarization models.

- We show that our methods have better or comparable performance to the current state-of-the-art trained on the entire training data (according to ROUGE), while requiring only 10% of the labelled data.

- We show that framing extractive summarization as a *question-answering problem* allows us to efficiently leverage large language models for weak supervision (e.g., GPT-3.5; Ouyang et al. 2022).

---

* Corresponding author n.mishra@amsterdamumc.nl. Work (partially) done during a research visit at ServiceNow.

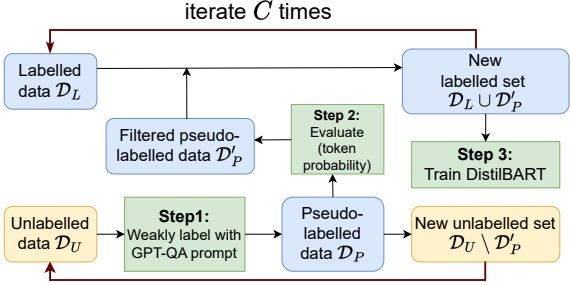

Figure 1: Overview of our approach.

**Related work**    Chen and Yang (2021) introduced Conversational Data Augmentation (CODA), a simple conversational data augmentation framework for *abstractive dialog summarization* with an optional extension for a semi-supervised learning setting (He et al., 2020; Xie et al., 2020). Sznajder et al. (2022) propose a heuristic-based weakly-supervised *abstractive summarization* model for the TWEETSUMM dataset. They first train separate models for customer and agent on a weakly-labeled set obtained by using the LONG and LEAD heuristics and then finetune such models on labelled conversation data. Liu and Lapata (2019) use pre-trained encoders like BERT and propose PreSumm, a model for *abstractive and extractive summarization* based on sentence classification.

## 2 Methodology

### 2.1 Notation

We assume a training dataset $\mathcal{D} = \mathcal{D}_L \cup \mathcal{D}_U$ with labelled ($\mathcal{D}_L$) and unlabelled examples ($\mathcal{D}_U$). Labelled examples $\mathcal{D}_L = \{u_i, s_i\}_{i=1}^{D_L}$ consist of a dialog instance $u_i = \{u_{i,1}, u_{i,2}, \cdots, u_{i,N}\}$ composed of $N$ input tokens and dialog $u_i$'s summary $s_i = \{s_{i,1}, s_{i,2}, \cdots, s_{i,M}\}$ with $M$ tokens. Unlabelled examples $\mathcal{D}_U = \{u_j\}_{j=1}^{D_U}$ consist of a dialog instance $u_j = \{u_{j,1}, u_{j,2}, \cdots, u_{j,K}\}$ with $K$ input tokens for which there is no summary. In most practical cases, $D_U \gg D_L$.

### 2.2 Iterative procedure for knowledge distillation

In Figure 1, we show an overview of our approach. We use an idea similar to teacher-student transfer learning(Sanh et al., 2020; Shleifer and Rush, 2020a; Tang and Huang, 2022) and propose an iterative training procedure with $C$ *cycles*. Each cycle of training consists of three steps: 1) Use a weak-labeller to generate pseudo-labels for dialogs $u_j \in \mathcal{D}_U$, 2) evaluate and select high-quality

pseudo-labelled examples ($\mathcal{D}_P$) for the next step without replacement, and 3) retrain summarization model using labelled and pseudo-labelled examples $\mathcal{D}_L^{new}$.

In practice, we pseudo-label all unlabelled samples $\mathcal{D}_U$ *only once* using GPT-3.5 and re-use the predicted labels at each cycle $c \in C$. We can do this efficiently for two reasons: the number of unlabelled samples in TWEETSUMM is small (i.e., 850 data points); and we do not fine-tune GPT-3.5, which is an efficient and performant alternative (as we will show in our experimental results).

**1) Weak labelling**    Before the first cycle ($c = 0$), we use GPT-3.5 (OpenAI, 2023) in a *few-shot setting* to generate pseudo-labels for all unlabelled examples $\mathcal{D}_U$. We refer to this set of pseudo-labelled examples as $\mathcal{D}_P = \{u_j, \hat{s}_j\}_{j=1}^{D_U}$. We framed the extractive summarization task as a question-answering problem for GPT-3.5. The dialogues were converted into a sequence of numbered sentences. The prompt consisted of an instruction that asked the model to return sentence numbers that adequately summarised the dialogue from both perspectives. We also provided several examples as context. We then extracted the sentences in the dialog corresponding to the numbers that the model returned to construct the pseudo-labeled summaries. This helped ensure that the pseudo-labels thus obtained were always extractive. Please refer to Appendix A for more details about the overall process of pseudo-labelling and samples of the prompts used.

**2) Evaluating and selecting pseudo-labelled examples**    We first evaluate the quality of each generated pseudo-labelled example in $\mathcal{D}_P$. We use the log-probability assigned by GPT-3.5 to tokens corresponding to the *numbers* of all sentences included as part of the summary in step 1, and sum these log-probabilities to compute a score for each pseudo-labelled example in $\mathcal{D}_P$. We then select a subset $\mathcal{D}_P' \subset \mathcal{D}_P$ of the $D_P'$ highest-scoring examples. Finally, we merge the original labelled data $\mathcal{D}_L$ with the selected pseudo-labelled data $\mathcal{D}_P'$, generating a new set of labelled examples $\mathcal{D}_L^{new} = \mathcal{D}_L \cup \mathcal{D}_P'$ to use to train the summarization model.

**3) Train summarization model using $\mathcal{D}_L^{new}$**    We train a sequence-to-sequence model on the updated set of labelled examples $\mathcal{D}_L^{new}$, which includes the original labelled examples $\mathcal{D}_L$ and the pseudo-labelled examples $\mathcal{D}_P'$ selected in step 2. We train

this model to generate an *extractive* summary $s_i$ for a dialog $u_i$ by minimising the negative log-likelihood $\mathcal{L}(\mathcal{S}) = \mathcal{L}(\mathcal{D}_\mathcal{L}) + \mathcal{L}(\mathcal{D}'_\mathcal{P})$, given that

$$\mathcal{L}(\mathcal{D}_\mathcal{L}) = \sum_{i=1}^{D_L} -\log p(s_{i,t}|s_{i,<t}, u_i; \theta),$$

$$\mathcal{L}(\mathcal{D}'_\mathcal{P}) = \sum_{j=1}^{D'_P} -\log p(\hat{s}_{j,t}|\hat{s}_{j,<t}, u_j; \theta),$$

where $s_{i,<t}$ ($\hat{s}_{j,<t}$) are the first $t-1$ tokens of the gold-standard summary $s_i$ (the generated summary $\hat{s}_j$).

**4) Convert generative summaries to Extractive**
The summarization model is given a dataset where the ground truth summaries always contain sentences from within the dialog, thus we hypothesize that the model should learn representations in a way to output exclusively extractive summaries. Even with qualitative and quantitative results validating this, there is no guarantee of an exclusively extractive output from a generative seq-to-seq model. Thus we add another step to ensure extractive summarization.

The generated summaries $s_i$ are converted to extractive summaries by a sentence matching procedure. Each sentence in the generated summary and the dialogue was first embedded using a pre-trained BERT-based sentence transformer model[1](Reimers and Gurevych, 2019). For each sentence embedding in the generated summary, we calculated its cosine distance with each sentence embedding from the original dialogue to obtain semantic textual similarity. We then replaced the summary sentences with the corresponding sentence in the dialog that had the highest similarity i.e. lowest cosine distance from them, constructing the final extractive summary.

## 3 Experimental Setup

**Data** In our experiments, we use the TWEET-SUMM dataset (Feigenblat et al., 2021), which contains $1,100$ labelled examples $\mathcal{D}_L$ of dialog between customers and customer support agent. We use the original splits and have 880, 110, and 110 examples for training, model selection, and testing, respectively. In semi-supervised experiments, we subsample a fraction of data as labelled and rest is used as unlabelled. We repeat these experiments

---

[1]We used the Sentence-Transformers library

with three random seeds and average results to account for variance. All the evaluations of extractive summarization for the models described in the experiments below are performed in a limited length of 80 tokens setting.

**Our Method** We used the method described in section 2.2. Concretely, we use Distil-BART (Shleifer and Rush, 2020b)—a distilled version of BART (Lewis et al., 2019) with 6 encoder and 6 decoder layers—as the sequence-to-sequence model. We set the number of examples $D'_P$ selected by the evaluator at each cycle as 16, the total number of cycles $C = 10$, and in each cycle we train the extractive summarization model for 10 epochs.

**Vanilla method** We also explore a 'vanilla' version of our method. Crucially, with this method we do not transfer knowledge from a massive LLM (e.g., GPT-3.5) into the summarization model but use the summarization model in a two-step noisy self-training paradigm(Chen et al., 2021).

### 3.1 Iterative training and GPT-3.5

We devise a number of baselines for this experiment. We use **LEAD-1** and **LEAD-2** (Kryscinski et al., 2019), two competitive extractive summarization baselines which take the first one (two) leading sentences each from the customer and from the agent as the final summary. We also use **LONG-1** where the agent's and the customer's *longest sentences* are taken as the final summary. These baselines are not iterative. We directly measure these heuristically generated summaries on the test set. We propose two iterative semi-supervised baselines where we use the agent's and the customer's *first sentence* (**LEAD-1-i**) and *longest sentence* (**LONG-1-i**), respectively, as the *initial summary* (§ 2.2, step 1), but instead of selecting the $X$ highest-scoring examples (§ 2.2, step 2), we randomly select $X$ examples to use as $\mathcal{D}'_P$ (§ 2.2, step 3).

### 3.2 Few-shot learning

For this set of experiments, we retrain state-of-the-art task-tuned language models for extractive dialog summarization. We use DistilBART fine-tuned on the XSum (Narayan et al., 2018, **DistilBART-xsum**) and CNN-dailymail (Nallapati et al., 2016, **DistilBART-cnn**) datasets, and further fine-tune these two models *once* using various subsamples of the TWEETSUMM dataset.

| Models | Iter. | R-1 | R-2 | R-L |
|---|---|---|---|---|
| **LEAD-1** | ✗ | 40.89 | 34.22 | 38.74 |
| **LEAD-2** | ✗ | 53.51 | 43.27 | 47.76 |
| **LONG-1** | ✗ | 54.47 | 46.56 | 50.58 |
| **LEAD-1-i** | ✓ | 62.64 | 53.13 | 57.46 |
| **LONG-1-i** | ✓ | 59.80 | 50.12 | 54.36 |
| **Ours (GPT-3.5)** | ✓ | **65.93** | **56.94** | **60.96** |

Table 1: Results on TWEETSUMM's test set. Comparison between five simple baselines and our method with GPT 3.5 as weak labeller and evaluator and DistilBART-cnn as summarizer. Baselines can be iterative (✓) or not (✗). **R-1**, **R-2**, and **R-L** correspond to ROUGE-1, ROUGE-2, and ROUGE-L F-measure, respectively.

## 3.3 Full data scenario

Here, we use state-of-the-art models for extractive dialog summarization—including models used in the original TWEETSUMM dataset paper (Feigenblat et al., 2021)—as well as instruction-tuned LLMs to directly summarize dialogs. **Pre-Summ** (Liu and Lapata, 2019) frames extractive summarization as a sentence classification problem by predicting whether sentences are noteworthy, and is the current SotA on TWEETSUMM for extractive summarization. In **GPT-3.5** we directly prompt the model to auto-regressively generate the entire summary for a dialog.

In **GPT-3.5-QA** we treat extractive summarization as a question-answering problem as we do in our method (§ 2.2),

We provide more details on how we prompt GPT-3.5 autoregressively and in a QA setting in Appendix A. For both GPT-3.5 and GPT-3.5-QA, we experiment in different few-shot in-context (Brown et al., 2020) settings– {0,1,2,4,8}–within the 4096 token limit. We measure the performance using Prompted Direct Inference(PDI), directly measuring the performance on test data.

## 4 Results

In § 4.1, we first discuss how our method compares to simple baselines (iterative and non-iterative), clearly showing it surpasses both. In § 4.2, we examine how our approach fares in a few-shot learning setting where there is no-to-little training data available. In § 4.3, we compare our best few-shot models with the current SotA using all training data available, and show that our method performs comparably to the current SotA while using only

| Models | Iter. | # labelled / unlabelled data points | | | |
|---|---|---|---|---|---|
| | | 0 / 850 | 1 / 849 | 8 / 842 | 80 / 770 |
| **D-BART-xsum** | ✗ | 3.93 | 33.51 | 44.56 | 47.16 |
| **D-BART-cnn** | ✗ | 30.95 | 42.87 | 46.65 | 50.75 |
| **Ours (vanilla)**‡ | ✓ | 50.72 | 50.16 | 51.42 | 51.83 |
| **Ours (GPT-3.5)**‡ | ✓ | **53.75** | **54.15** | **55.82** | **56.94** |

Table 2: ROUGE-2 F-measures on TWEETSUMM's test set. **D-BART** is DistilBART. ‡: Our methods either only use DistilBART-cnn as weak labeller and summarizer (vanilla) or use GPT-3.5 as weak labeller and DistilBART-cnn as summarizer (for details, see § 2.2).

10% of the training data. We use the **ROUGE-1**, **ROUGE-2** and **ROUGE-L** metrics (Lin, 2004) since these are widely used metrics to automatically evaluate text summarization models, and also for comparing our results with existing work.

### 4.1 Iterative training

We report results in Table 1. We observe that Our approach significantly outperforms baselines directly using first and longest sentences from the customer and agent as the summary in both iterative and non-iterative scenario across all metrics. Moreover, cyclic baselines clearly outperform their non-cyclic versions, highlighting the importance of distilling knowledge into the summarization model in small steps.

### 4.2 Few-shot learning

In Table 2, we first observe consistent gains from using semi-supervised learning and iteratively training the summarization model (DistilBART) using more and more pseudo-labelled dialogs, over the same models fine-tuned on only labelled data. We also consistently improve DistilBART's performance by using GPT-3.5 as the weak labeller and evaluator, i.e., between 3.3%–5.1% increase in ROUGE-2 compared to our vanilla method.

### 4.3 Full data scenario

In Table 3, we show that our best model using GPT-3.5—trained on only 10% of the data—outperforms PreSumm—which is trained on the entire training dataset—according to ROUGE-1 and ROUGE-2, and is only 3.5% behind according to ROUGE-L. We also outperform DistillBART models finetuned on the entire dataset by almost 2 points across all metrics using only 10% of the labelled data. Among the in-context prompt based models, GPT-3.5-QA outperforms GPT-3.5 which is notable as it shows why we used it for pseudo-labelling. The

| Models | Iter. | Semi. | R-1 | R-2 | R-L |
|---|---|---|---|---|---|
| **GPT 3.5** | ✗ | in-ctx | 58.11 | 49.48 | 54.86 |
| **GPT 3.5-QA** | ✗ | in-ctx | 62.05 | 52.98 | 57.63 |
| **Full data** | | | | | |
| **D-BART-xsum** | ✗ | ✗ | 61.24 | 54.05 | 58.10 |
| **D-BART-cnn** | ✗ | ✗ | 63.61 | 54.84 | 58.91 |
| **PreSumm**[†] | ✓ | ✗ | 65.16 | 55.81 | **64.37** |
| **10% of training data** | | | | | |
| **Ours (vanilla)**[‡] | ✓ | ✓ | 61.30 | 51.83 | 56.89 |
| **Ours (GPT-3.5)**[‡] | ✓ | ✓ | **65.93** | **56.94** | 60.96 |

Table 3: Results on TWEETSUMM's test set. **D-BART** is DistilBART. [†]: PreSumm is the model proposed in Liu and Lapata (2019). [‡]: These are our best models in each case, which is always the corresponding model trained on 10% of the training data. **R-1**, **R-2**, and **R-L** correspond to ROUGE-1, ROUGE-2, and ROUGE-L F-measure, respectively.

scores for GPT-3.5 and GPT-3.5-QA are from two-shot in-context learning which outperformed zero, one, four and eight shot performances.

## 5 Conclusions and Future Work

In this work, we introduce a semi-supervised approach for extractive summarization to distill knowledge from general-purpose LLMs into smaller specialized summarization models. We demonstrate our method on the TWEETSUMM dataset, and show that while training on only 10% of the data our method is competitive to PreSumm, the current state-of-the-art, which uses 100% of the training data. We outperform PreSumm by 0.3% according to ROUGE-1 and ROUGE-2, but are still 4% behind according to ROUGE-L. As future work, we will extend our method to other text summarization datasets. We will also explore using various open-source, tunable LLMs for both summarization and pseudolabelling.

## Limitations

We show promising results achieved using instruction-tuned LLMs to help with semi-supervised dialog summarization, But there are a few limitations to the work that are laid out in this section. One of the major issues is in terms of breadth of the experiments. The current work deals with only one dataset i.e TweetSumm and one kind of summarization, Extractive dialogue summarization. Further experiments involving other popular datasets comprising different domains, sizes and

tasks will help consolidate the performance of our method.

Another significant limitation is the closed source nature of the model we are using for pseudo-labelling i.e GPT-3.5(text-davinci-003). This makes it prohibitively expensive to fine-tune or reproduce in cases where the number of samples to be pseudo-labelled is substantial. Therefore we ended up using a fixed teacher model and generating the pseudo-labels only once. Having a more tunable, open-source instruction-tuned LLM as the weak labeller would have enabled prompt-learning to further improve pseudo-labels. This might have helped us unlock other aspects of teacher-student learning by allowing feedback from the smaller student model to improve/align the larger model updating the pseudo-labels every cycle i.e meta pseudo-labeling (Pham et al., 2021). It would also have provided more insight into the method while being less expensive for our task. We tried to overcome this limitation by exploring open-source instruction tuned models like alpaca-7B, vicuna-13B as our weak labeller. But the compute requirements to host and infer with these models, inferior and slow inference using the low parameter and quantized versions proved that they are not yet tractable and robust.

## Ethics Statement

The dataset we used, TWEETSUMM is constructed from an open-source customer support dataset available here .The data, although from real public interaction, has been anonymized. It's annotation was crowdsourced. Further we use GPT-3.5 for pseudo-labelling.Since this is from real-world customer-agent interactions we can, in rare cases and based on context get spurious, undesirable or biased outputs. Although chances of that are minimized by framing it into a QA problem and limiting the generated tokens to a very small window.

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

# A   Appendix

## A.1   Pseudo-labelling using GPT

We used instruction tuned GPT-3.5(text-davinci-003) for Prompt Guided Unlabeled data annotation for extractive summarization. We explore two different approaches to the problem that are described below. We test and report results for the two methods through **Prompted Direct Inference(PDI)**, directly measuring the performance by annotating test data.

### A.1.1   GPT-3.5 QA

Here we frame the task of weak labelling as a *question-answering problem*: we first split a given dialog $u_j$ into sentences using a *sentence splitter*;[2] we then number these sentences from 1 until $N$, where $N$ is the total number of sentences found in $u_j$ (e.g., if the original sentence reads *'This is a sentence.'* and it is the 2nd sentence in the dialog, it becomes *'2) This is a sentence.'*); In order to convert the summary to the suitable format, we first compare the sentences in summary to the sentences in dialogue to find their positions in the dialog and then map the summary into a fixed structured sentence that highlights the numbers of the sentence containing the summary from both perspectives.

We build a prompt containing a few labelled examples (i.e., gold-standard dialog and summary) as context followed by the dialog $u_j$ for which we wish to generate a summary; we prompt GPT-3.5 asking it *for the numbers* of all the sentences that must be included as part of the summary. To be more precise, we ask it to answer with sentence numbers that describe the issue being faced by the customer in the dialog and sentence numbers that best highlight the answers provided by the agent.

Once we obtain the numbers of the sentences GPT-3.5 deems suitable to be part of the summary, we build the summary $\hat{s}_j$ by simply extracting these sentences from the original input $u_j$.

Figure 2 shows an example of a prompt we built for pseudolabelling along with the response from GPT-3.5.

### A.1.2   GPT-3.5

In **GPT-3.5** we directly prompt the model to auto-regressively generate the entire summary for a dialog. We provide labeled pairs of dialogue and summary as context and ask it to generate summary for the next dialogue. Since this is an extractive

---

[2]We used the spaCy sentencizer for this

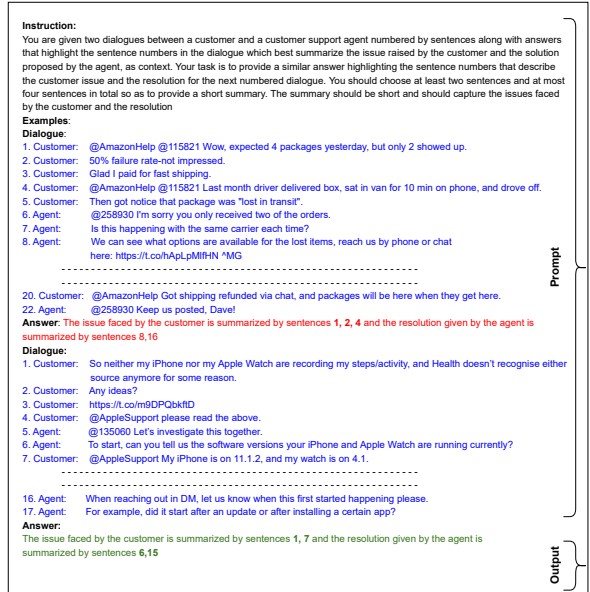

Figure 2: The prompt used for **GPT-3.5 QA in-context extractive dialog summarization and pseudo-labelling**. The first part contains the prompt with instructions and examples, and the second part is the response generated. Blue represents the dialogs, red represents the sample summaries/answers part of the prompt and green represents the final summary/answer generated. We show only one shot in-context here for brevity

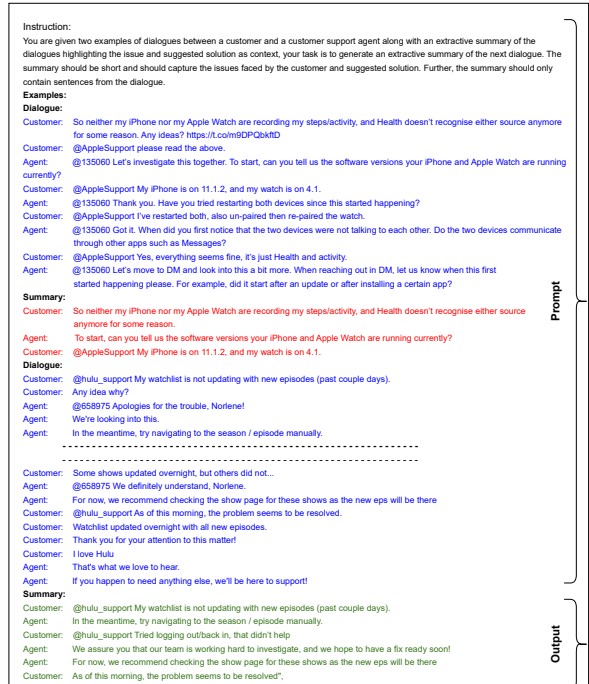

Figure 3: **The prompt used for GPT-3.5 in-context extractive dialog summarization.** The first part contains the prompt with instructions and examples, and the second part is the response generated. Blue represents the dialogs, red represents the sample summaries part of the prompt and green represents the final summary generated. We show only one shot in-context here for brevity

summarization problem, we make sure to clarify in the prompt instruction to use sentences that are part of the dialog and not to synthesize new sentences or paraphrase. Figure 3 shows the prompt that we used for getting summaries with GPT-3.5 by completion.

# B    Test learning curves

Figure 4 shows the test learning curves for our summarization model(distillBART-cnn) across 10 cycles of semi-supervised training. Each curve represents different settings in terms of numbers of labelled samples at the start. Every cycle, we add new pseudo-labelled samples generated with GPT-3.5 QA.

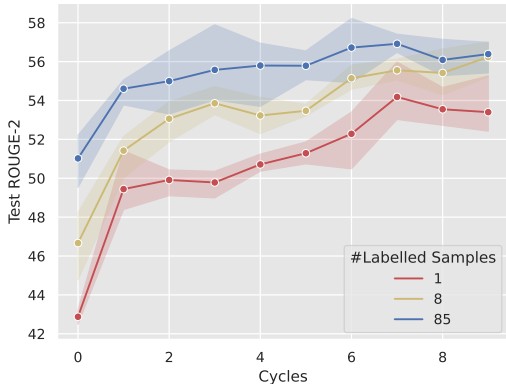

Figure 4: Test curves for our method (using GPT-3.5 as weak labeller and evaluator) when trained on 1, 8, and 85 samples (i.e., approximately 0.1%, 1%, and 10% of the TWEETSUMM training data).