# OpenReview forum: "LLM aided semi-supervision for efficient Extractive Dialog Summarization"
_EMNLP/2023/Conference — EMNLP 2023 Findings_

### Official Review · Reviewer_P49m · 2023-07-24

**Soundness:** 4

**Excitement:**

4: Strong: This paper deepens the understanding of some phenomenon or lowers the barriers to an existing research direction.

**Paper Topic And Main Contributions:**

This paper proposes a new method that can effectively use unlabeled data to generate high-quality summaries for chat dialogs. This method frames summarization as a question-answering problems and use a large language model to generate summaries for unlabeled data to form pseudo labeled data and fine-tunes the summarization model to generate summaries using pseudo labeled data. Overall, the model achieved results similar to state-of-the-art models using only 10% of data.

**Reasons To Accept:**

This paper proposes a new method of efficiently using unlabeled data on chat dialogs summarization task and it is hopefully popularized to other tasks which have a need for a large number of labeled data.

**Reasons To Reject:**

The method proposed by this paper uses a closed source model to form pseudo labeled data, which means it may be difficult to fine-tuning this model for a higher quality of pseudo labeled data. And experiments using this method on open-source models have not been successful, making it difficult to prove that this method can be extended to other language processing tasks.

**Reproducibility:**

3: Could reproduce the results with some difficulty. The settings of parameters are underspecified or subjectively determined; the training/evaluation data are not widely available.

**Reviewer Confidence:**

5: Positive that my evaluation is correct. I read the paper very carefully and I am very familiar with related work.

---

> ### Author Rebuttal · Authors · 2023-08-28
>
> We would first like to thank the reviewer for their insightful comments, and for finding our work interesting.
>
> We agree with the reviewer's point regarding the use of a closed-source model. Not being able to use an open-source model largely has to do with logistic bottlenecks such as lack of computing resources on our end, and stable and accurate open-source models.
>
> With open-source models increasingly inching closer to the performance of closed-source models on a range of tasks and benchmarks, the claims and results of our paper should be reproducible with open-source models (e.g., Llama-2). They will also provide an additional advantage of low cost and the ability to fine-tune the models or prompts to generate higher quality pseudo labels every cycle as mentioned by the reviewer, thus in fact leading to better results.

---

### Official Review · Reviewer_2gVZ · 2023-08-03

**Soundness:** 4

**Excitement:**

2: Mediocre: This paper makes marginal contributions (vs non-contemporaneous work), so I would rather not see it in the conference.

**Paper Topic And Main Contributions:**

This paper investigates the use of an instruct-tuned model, GPT3.5 for data augmentation in an extractive dialogue summarization setting. The authors propose to automatically obtain summary sentences from dialogue by GPT3.5. Then a log likelihood-based filtering method is applied to filter instances with low confidence. Finally, the synthesized summary sentences are used for training a generative summarizer.

As a strength, The experiments show that the performance of the model trained on the proposed pseudo dataset is comparable to the model trained on the human-authored dataset.

This paper is easy to follow, however, some notations in Sec. 2.1 does seem not clear (see the questions for authors.).
Although the authors claim that they focus on the extractive setting, the trained model is not guaranteed to be an extractive model (see the weakness below.).
Overall, this paper proposes a straightforward usage of GPT3.5 and comprehensive evaluations provide insights for the community, However, the proposed idea is a simple replacement of existing data augmentation techniques by a model with few-shot ability, which may provides less novelty.

**Questions For The Authors:**

- Are the notations in Sec 2.1 correct? I assume that gold summaries are given to a sequence of utterances, but the notations in the paper assume that a summary is given to each utterance (u_{i}).

- In the equation written in line160-161, this modelling is for abstractive summarization not for extractive, thus, this model cannot guarantee extractive output. If my understanding is wrong, please point it out.

**Reasons To Accept:**

- This paper is well-written and easy to follow.
- The experiments show that the performance of the model trained on the proposed pseudo dataset is comparable to the model trained on the human-authored dataset.
- Straightforward and reasonable usage of GPT3.5.

**Reasons To Reject:**

- The synthesized dataset is not guranteed for a purely extractive setting.
 (i.e., the authors used ``You should choose at least two sentences and at most four sentences in total'' for the prompt, but GPT3.5 is a language generator, thus, the produced dataset is not guaranteed to contain only extracted sentences.)

- The authors trained a generative model although the authors focus on an extractive setting. This modeling does not guarantee  the output summary is extractive.

**Reproducibility:**

4: Could mostly reproduce the results, but there may be some variation because of sample variance or minor variations in their interpretation of the protocol or method.

**Reviewer Confidence:**

4: Quite sure. I tried to check the important points carefully. It's unlikely, though conceivable, that I missed something that should affect my ratings.

**Typos Grammar Style And Presentation Improvements:**

- \bf{Related Work} -> \section{Related Work}.

---

> ### Author Rebuttal · Authors · 2023-08-29
>
> We are thankful to the reviewer for their insightful review, we also acknowledge the reviewer's detailed explanation and close reading, highlighting notational and style issues. We elaborate on some of the points raised below.
>
> **The synthesized dataset is not guaranteed for a purely extractive setting**: As described in section 2.2 (lines 134-136)  and demonstrated in appendix A.1.1, a major ingenuity of the work is in how we guarantee that the GPT pseudo labels are extractive by mapping the summarization as a QA task making it generate sentence numbers that it "thinks" should describe the issues and solutions. We then use these numbers to extract the specific sentences thus forming our summary. Also, we observed zero cases during testing where the model generated hallucinations/sentence numbers outside the document size, but we had fallbacks in place to handle those, such as discarding unrealistic sentence numbers and falling back on the LEAD heuristic in case of malfunction. We include a discussion in how we handle these exceptions, were they to happen, in the camera ready if our paper is accepted for publication.
>
> **The authors trained a generative model although the authors focus on an extractive setting. This modeling does not guarantee the output summary is extractive.**
> * Generative models such as BART-large performed comparably to encoder-based classifier models for extractive summarization in our experiments,
> * Thus, our hypothesis was that given a dataset where the ground truth summaries always contain sentences from within the document, the model will learn representations in a way to output extractive summaries.
> * We obtained both qualitative and quantitative results to highlight it.
> * _However, the point raised by the reviewer regarding the lack of such a guarantee is still valid_ and we added an additional step to ensure am extractive output. We took the output from the BART model and performed a per-sentence match with the dialog using sentence transformers dot score, to extract the closest matching sentence from the dialog to make sure it's always extractive. Adding this additional step guarantees extractive summarization.
> * As an ablation, we shall be adding experiments with PreSumm, which is an extractive model as the summarizer instead of BART in the final version of the paper.
>
> **On the Questions**
> * Yes the notations are correct, but I think our wording is confusing, by utterance (u_{i}) here, we mean a dialogue i.e. set of utterances. We shall fix it, thanks for the catch.
> * Based on the above point, we will add an additional step to represent matching on top of abstractive summarization in the equation, along with a section on preSumm.
>
> **Regarding the point about novelty**, we might not have been clear but would like to stress that this is not merely a data augmentation technique. We show here that knowledge from generic high-parameter models trained on billions of tokens can serve as a useful starting point for smaller specialized models distilling their knowledge before finetuning in a very specific use-case thus helping in low-data and no-data scenarios. We also show the effectiveness of these models as a good substitute for human annotators, lowering costs while improving performance. It also lays the foundation for an incremental semi-supervised method for summarization, which can be extended to using open-source models, enabling even better results through fine-tuning, and improving the quality of pseudo-labels every cycle.

---

### Official Review · Reviewer_Bx3Q · 2023-08-09

**Typos Grammar Style And Presentation Improvements:** 1. Lines 326 & 532, text-davinci-003 …
**Soundness:** 4

**Excitement:**

2: Mediocre: This paper makes marginal contributions (vs non-contemporaneous work), so I would rather not see it in the conference.

**Paper Topic And Main Contributions:**

This paper proposes a method for extractive summarization of customer-agent dialogs, using LLMs to generate pseudo-labels for a dialog. The main contribution is a novel use of pseudo-labels for semi-supervised learning, which allows for the transfer of knowledge from the large LLM into a smaller specialized model. The paper demonstrates the effectiveness of this method on the TWEETSUMM dataset and shows that it achieves a good performance while using 10% of the labeled data.

**Questions For The Authors:**

1. Is there an oracle result for this TWEETSUMM dataset?
2. Have the authors conducted the experiment that: 1) uses GPT-3.5 as the labeler and evaluator, and the PreSumm as the summarizer; 2) Use PreSumm as the labeler and summarizer (vanilla)? This is concerned as a missing part of the ablation study.
3. Case Study is needed.

**Reasons To Accept:**

1. The paper proposes a novel approach to extractive summarization using LLMs.

2. The paper is easy to follow.

2. The proposed method demonstrates its effectiveness on the TWEETSUMM dataset while using 10% of the labeled data.

**Reasons To Reject:**

1. The method is not efficient enough. It needs to train the model for ten iterations, and in each iteration, there are ten epochs of training. In Table 3, using GPT-3.5 as the labeler and evaluator does not largely improve the performance; this requires further analysis.
2. Writing could be improved, having some typos and format issues. I am doubtful that the authors wrongly used GPT-3 API (text-davinci-003 is claimed in the paper) rather than GPT-3.5 (gpt-3.5-turbo)
3. GPT-3.5 is a generative model, which does not guarantee extractive summarization.

**Reproducibility:**

4: Could mostly reproduce the results, but there may be some variation because of sample variance or minor variations in their interpretation of the protocol or method.

**Reviewer Confidence:**

4: Quite sure. I tried to check the important points carefully. It's unlikely, though conceivable, that I missed something that should affect my ratings.

---

> ### Author Rebuttal · Authors · 2023-08-29
>
> We are grateful to the reviewer for his insightful comments and questions. We have tried to address specific concerns and questions in a concise manner below
>
> **The method is not efficient enough**
> Our method is focused on data efficiency, as we try to replicate full data performance using only a fraction of it while maintaining the training efficiency.
> Having said that since we are incrementally adding examples for training every cycle by selecting the most confident pseudo-samples and finetuning the summarizer in every cycle on the new composite dataset, it is important to train it for several epochs for it to learn from the samples and converge. It is a standard process while training any generative models **[2]**, as well as semi-supervised learning **[1]**.
>
> **‘Using GPT-3.5 as the labeler and evaluator does not largely improve, requires further analysis’**, It’s not clear what the improvement is compared against.  In Table 3 we show that our method using GPT as a weak labeler and evaluator improves upon the state of the art(PreSumm) on the tweetsumm Dataset using only 10% of the annotated dataset, which is a significant improvement as you rightly noted in the main contributions. It is also ~4 points higher than the BART semi-supervised model.
>
> **Authors wrongly used GPT-3 API (text-davinci-003 is claimed in the paper) rather than GPT-3.5 (gpt-3.5-turbo)**
> * Text-davinci-003 belongs to the instructGPT family of models which is an instruction-tuned iteration of GPT-3. It is thus a part of the GPT-3.5 family of APIs as can be verified from the openAI page **[3]**, while davinci is a part of GPT3.
> * Gpt-3.5-turbo is a chat-optimized version of it. We used text-davinci-003 because it consistently outperformed gpt-3.5-turbo in a few shot settings using prompted direct inference.
> * Also, it provides us with token probabilities that we use to get a confidence score as explained in section 2.2.2, which is not available for gpt-3.5-turbo
>
> **GPT-3.5 is a generative model, which does not guarantee extractive summarization.** As described in section 2.2(lines 134-136)  and demonstrated in appendix A.1.1, a major ingenuity of the work is in how we guarantee that the GPT pseudo labels are extractive by mapping the summarization as a QA task making it generate sentence numbers that it feels describe the issues and solutions. We then use these numbers to extract the specific sentences thus forming our summary. Also, we observed zero cases during testing where the model generated hallucinations/sentence numbers outside the document size, but we had fallbacks in place to handle those, such as discarding unrealistic sentence numbers and falling back on the LEAD heuristic in case of malfunction.
>
> **On the Questions**
> * To the best of our knowledge, and based on an exhaustive literature review around this dataset, there doesn’t exist an oracle score for TweetSumm.
> * Tuning parameters for GPT-3.5(temperature 0.2, top_p=1, max_tokens=48 with a logit bias for numeric tokens)  and distillbart will be added.
> * Adapting the available implementation of PreSumm for tweetsumm dataset is non-trivial, and given the limited scope of a short paper, we thus only used it as a baseline. But the suggested experiments by the reviewer are indeed interesting ablations that we will add to the final version of the paper.
> * It is unclear what is meant here by case study.
>
> **References**
>
> [1] Jiaao Chen and Diyi Yang. 2021. Simple conversational 376 data augmentation for semi-supervised abstractive 377 dialogue summarization. In Proceedings of the 2021 378 Conference on Empirical Methods in Natural Lan- 379 guage Processing, pages 6605–6616.
>
> [2] Sam Shleifer and Alexander M Rush. 2020b. Pre- 492 trained summarization distillation. arXiv preprint 493 arXiv:2010.13002
>
> [3] https://platform.openai.com/docs/models/gpt-3-5

---

### Meta-Review · Area_Chair_SdT1 · 2023-09-18

**Recommendation:** 2

**Metareview:**

The three papers propose methods for extractive summarization of customer-agent dialogs using large language models (LLMs) to generate pseudo-labels for dialog data. The paper's main contribution is the novel use of pseudo-labels for semi-supervised learning, which enables knowledge transfer from the LLMs to specialized models. They demonstrate the effectiveness of their methods on different datasets, achieving good performance with minimal labeled data. However, several concerns about the efficiency of the methods, and uncertainty about guaranteeing extractive summarization in some cases need to be addressed. Overall, these papers propose innovative techniques for dialogue summarization, but certain limitations and clarity issues require further revision.

---

### Decision · Program_Chairs · 2023-10-07

**Decision:**

Accept-Findings

**Comment:**

The three papers propose methods for extractive summarization of customer-agent dialogs using large language models (LLMs) to generate pseudo-labels for dialog data. The paper's main contribution is the novel use of pseudo-labels for semi-supervised learning, which enables knowledge transfer from the LLMs to specialized models. They demonstrate the effectiveness of their methods on different datasets, achieving good performance with minimal labeled data. However, several concerns about the efficiency of the methods, and uncertainty about guaranteeing extractive summarization in some cases need to be addressed. Overall, these papers propose innovative techniques for dialogue summarization, but certain limitations and clarity issues require further revision.